# Student Teachers' Classroom Impact during Their Practicum in the Times of the Pandemic

Laura Alonso-Díaz [1],*[ID], Gemma Delicado-Puerto [2], Francisco Ramos [3] and Cristina Manchado-Nieto [4]

1   Education Department, Universidad de Extremadura, Avda. Universidad s/n, 10071 Cáceres, Spain
2   English Philology Department, Universidad de Extremadura, Avda. Universidad s/n, 10071 Cáceres, Spain
3   Department of Teaching and Learning, Loyola Marymount University, 1 LMU Drive 2649, Los Angeles, CA 90045, USA
4   Department of Didactics of Language and Literature, Universidad de Extremadura, Avda. Universidad s/n, 10071 Cáceres, Spain
*   Correspondence: laulonso@unex.es

**Abstract:** The COVID-19 pandemic impacted societal structures worldwide. In the educational realm, the forced closure, and subsequent reopening of school settings disrupted the personal and professional lives of administrators, teachers, parents, and students. Two groups of Spanish stakeholders affected by the return to face-to-face instruction during the pandemic were the University of Extremadura's student teachers and their mentors during the practicum, as student teaching is known in Spain. This study investigated 28 student teachers' and 26 mentors' responses to a questionnaire enquiring on the main challenges resulting from the pandemic, as well as student teachers' contributions to the classroom during this difficult time. Cualitative software was used to analyze participants' responses allowed us to identify four main themes: attitudes; classroom management issues; adaptations and restrictions; and academic–experiential modifications. Overall, the findings revealed student teachers' and mentors' positive opinions about their joint work experience and about student teachers' logistical and technological support.

**Keywords:** student teaching; practicum; teacher training; mentor teachers; COVID-19; educational technology

## 1. Introduction

The sudden, unexpected outbreak of the COVID-19 virus has impacted school systems worldwide in a way no individual or institution could have ever predicted before [1–3]. The pandemic forced educational institutions around the world to implement drastic restrictions affecting schools and classrooms that frequently left administrators, teachers, and students wondering whether and when they would be able to return to a certain degree of personal and professional normalcy in their lives [4–6].

Similarly to so many countries worldwide (See, for example, the two monographs of the Revista Iberoamericana de Educación on Educación y pandemia published in 2021), educational authorities in Spain saw themselves rushing to draft ever-evolving directives on the adoption of health-related measures and alternative modes of instructional delivery. Pending fluctuating numbers of contagions allowing or restricting teachers' and students' returns to their classrooms, numerous teachers in the country struggled as they attempted to learn about, and use, available technological resources to conduct asynchronous and hybrid sessions remotely. In this dramatic context, some of the most fortunate ones received unexpected amounts of help from the very student teachers they were to mentor during the practicum, the student teaching component of the B.A. degree in Education in Spain.

### 1.1. The Pandemic

The virus spread rapidly in Spain and within 2 months, it had affected the 17 autonomous regions and 50 provinces of the country [7]. In view of the exponential number of

contagions and deadly cases growing uncontrollably, the Spanish government sanctioned a countrywide lockdown, one of the strictest in the world. The lockdown imposed the closing down of schools and universities for four months. After weeks of never-before-seen restrictions, the State of Alarm ended on 21 June 2021; severe outdoor regulations and rigorous health measures finally ended (Health-related areas or spaces and public transportation still require masks). In September 2020, schools and most universities reopened again but restrictions lasted for the whole school year 2021–2022 [8].

### 1.2. The Reopening

Most Spanish schools and universities were authorized to reopen in September of 2020, offering in-person classes under strict health-related measures (frequent hand washing, airing classrooms, and mandatory mask-wearing), although the back-and-forth with asynchronous and hybrid classes continued, contingent upon the spread of the pandemic. While the virus appeared to be under control in a majority of educational settings due to the consistent implementation of safety protocols [9,10], one adopted measure, the incorporation of remote modes of instruction, raised the issue of teachers' knowledge of remote modes of instruction [11], and schools and families' potential access to digital tools. In regards to the latter, Sanz et al. [12] found that, "there was no divergence in access to digital educational resources due to the level of family income, at least with regard to free access digital educational resources" (p. 2). However, reports by Beaunoyer et al. [13], Berman [14], UNICEF [15], or UNESCO [16] highlighted the heavier impact of the pandemic on low-income families due to the uneven access to and distribution of the necessary technological devices required to facilitate remote learning.

Moreover, other reports evidenced the lack of preparedness of the educational system to support teachers' and families' technological needs during this time. Unfortunately, the COVID-19 crisis revealed the necessity for teachers to have digital skills in order to effectively teach online. Teachers should be able to exploit, use, and apply digital technologies in all educational activities [17,18].

Ironically, notwithstanding the tragedies brought about by the pandemic, UNICEF saw it as a silver lining to prevent potentially similar collapses in the future: "it will be useful both in times of normality and crisis to build teachers' capacity to manage a remote virtual classroom, improve their presentation techniques, train them to tailor follow-up sessions with caregivers and blend technology effectively into their lessons" [19].

Spanish teachers faced dire situations during the lockdown and the State of Alarm mentioned above, given the uncertainties caused by endlessly evolving COVID scenarios. Similarly to so many thousands of colleagues around the world, they experienced stress, fatigue leading to exhaustion, depression, and difficulties with newly adopted pedagogical tools for both remote and face-to-face instruction [20–26]. Fortunately for some of them, however, unexpected amounts of logistical and technological help came from novice student teachers during the practicum.

### 1.3. The Practicum

Student teaching, or the practicum, as it is known in Spain, is a collaborative effort between Spanish elementary schools and universities' Colleges of Education, with the purpose of familiarizing student teachers with their future profession. The time spent in real classrooms is expected to expose student teachers to the realities of school settings, provide them the opportunity to observe and evaluate the implementation of effective learning strategies and techniques, and help them fine tune their own lesson preparation and delivery routines [27–29]. Therefore, the overarching component of the practicum is the development of student teachers' capacity to reflect on aspects of the application of theory into practice and on their own ability to deliver instruction in the classroom [30]. For these reasons, the practicum is considered the backbone of the B.A. degree in Education and a crucial stepping stone in student teachers' introduction to the profession [31].

Royal Decree 592/2014 of 11 July 2014 [32], established two sections of the practicum, practicum I and practicum II, in child and elementary education, which are held, respectively, during the 3rd and the 8th semester of the B.A., in public, private, or semi-public schools (private schools receiving public funds). To broaden their experiences, student teachers are expected to complete the practicum in two different settings, at least one of which must be a public school. The 1200 total contact hours of the practicum, equivalent to 24 ECTS (European Credit Transfer and Accumulation System) credits, are divided into two equal periods of 600 h each. Upon completion of this time, student teachers will have spent 750 h in schools, attended 12 face-to-face seminars totaling 50 h, and completed 400 h of independent work.

University of Extremadura (UEx from now onwards) student teachers enrolled in practicum I conduct observations in real classrooms and lead guided and independent teaching practices under the supervision of a mentor teacher. In practicum II, they add observations in their respective specialization, i.e., foreign languages, music, physical education, or general education. Their personal reflections during this time constitute the basis of the seminars they must attend during both sections of the practicum.

In order to place student teachers in participating schools, the Teachers' College at the UEx publishes a list of teachers in schools who are willing to mentor prospective student teachers. The college subsequently ranks student teachers per their academic grades, so that those with the highest grades have preference at the time of selecting schools. Once in their school placements, student teachers are jointly supervised by one of their degree professors and their appointed school mentor. Professors' duties include guiding and mentoring student teachers, closely working with mentors, and helping student teachers increase their repertoire of resources. Mentors, for their part, commit to hosting student teachers, facilitating their integration into their schools, familiarizing them with school practices per the established Faculty Traineeship Plan guidelines, and advising them on effective pedagogical and didactic issues. In sharing valuable knowledge about the profession, mentors thereby become student teachers' models and references [30,33] and contribute to turning schools into spaces for learning, innovation, and research [29]. Mentors feel this cooperation is a way of giving back to the system for the guidance they received during their pre-service training. While not awarded monetary stipends for their work with student teachers, they are granted additional points should they decide to request a transfer to another school.

In the context of the COVID-19 pandemic, the research question that aims to feed this area of study is "How did student teachers from the University of Extremadura contributed to their mentors and school contexts during the reopening of school centers after the confinement period of the COVID-19 pandemic?" The objective of this research thereby intended to identify the main challenges that arose in the school centers of Extremadura (Spain) during the post-confinement pandemic scenario, and student teachers' contributions to the classroom during that time. The following pages describe the methodology used, the main findings of the study, and conclude with a discussion and recommendations for further research.

## 2. Methodology

### 2.1. Study Design and Data Collection

Document analysis using ATLAS.ti software was used to examine the data gathered in the study, given that the research focuses on analyzing the comments participants produced in a written format [34]. Such comments resulted from individual perceptions based on particular experiences in a specific context [35]. The researchers coded in stepwise or cyclical procedures until achieving saturation [36], and subsequently categorized the data into general themes. Data analysis took place during the months of June and July of 2021.

UEx's-required research protocols were followed prior to, and during, the data gathering process. The researchers contacted the pre-selected mentors and student teachers via Google forms to make them aware of the purposes of the project, inquire on their interest in

responding to the questionnaires and, upon their acceptance, sign and return the required consent form, as well as their respective questionnaire. In addition to summarizing the goal of the project, the consent form informed participants that they would not be identified by name, that responses would be presented in group format, and that individual comments would just be identified by gender.

### 2.2. Participants

The study was carried out between the months of February and May, 2021, a time during which schools in Spain had reopened and student teachers were thereby able to enroll in practicum I. Of the 248 total student teachers and as many mentors engaged in the practicum, the authors selected 28 student teachers and 26 mentors representing a wide range of specializations, schools, and programs as participants in the study. Participants therefore constituted a sample of convenience representing the different schools (urban, rural, semiprivate, etc.) present in the Spanish system and all majors or itineraries of specialization (early childhood education, primary education, foreign languages (English or French), physical education, general education, and music). The mentors selected had a broad experience as educators, ranging from 3 to 38 years, as well as mentoring student teachers (from 1 to 25 years). Table 1 presents data on the total number of student teachers and mentors involved in the practicum, and on the numbers of student teachers and mentors participating in the project [37].

**Table 1.** Participants.

| School Types and Majors | Total Number of Student Teachers | Total Number of Mentors | Selected Student Teachers | Selected Mentors |
|---|---|---|---|---|
| **Elementary public schools** | **103** | **103** | **17** | **10** |
| Early Childhood Education | 53 | 53 | 6 | 4 |
| General Education | 7 | 7 | 2 | 1 |
| Music | 14 | 14 | 4 | 2 |
| English | 29 | 29 | 5 | 3 |
| **Elementary Semi-public schools** | **142** | **142** | **8** | **13** |
| Early Childhood Education | 60 | 60 | 3 | 6 |
| General Education | 17 | 17 | 2 | 5 |
| Physical Education | 38 | 38 | 1 | 1 |
| English | 27 | 27 | 2 | 1 |
| **Elementary Rural public schools** | **3** | **3** | **3** | **3** |
| Early Childhood Education | 1 | 1 | 1 | 1 |
| General Education | 1 | 1 | 1 | 1 |
| English | 1 | 1 | 1 | 1 |
| **Totals** | **248** | **248** | **28** | **26** |

### 2.3. Instruments

Two questionnaires, one for the mentors (Appendix A) and another one for the student teachers (Appendix B), were developed by the authors. They both consisted of nine questions divided into two sections. The first one gathered demographic data from the participants; the second one, consisting of six open-ended questions, enquired on mentors' and student teachers' opinions about surging classroom challenges during the pandemic, student teachers' contributions to logistical and organizational classroom routines during the same time, and overall evaluations of the experience. Small differences in the phrasing of said questions in each questionnaire (i.e, "As a student teacher at your school, what

were your main challenges during the pandemic?" v. "As a mentor teacher at your school, what were your main challenges during the pandemic?") intended to facilitate comparisons among mentors' and student teachers' responses.

### 2.4. Data Analysis

For Taylor and Bogdan [38], qualitative investigations revolve around descriptive data such as individuals' written and orally expressed or observable behaviors. This approach permits us to obtain a holistic vision of problems and data through empathetic understanding which, in turn, allows us to explain how individuals navigate daily situations affecting them [39]. The present study uses this approach, which allows us to analyze mentors' and student teachers' answers to questions about vital experiences and behaviors.

Quality criteria are integral components of research projects intending to demonstrate credibility, transferability, dependability, and confirmability [40]. The present project addresses: credibility (trust in the veracity of the findings) by including a wide representation of participants in similar contextual situations; transferability (applicability to additional contexts) by investigating issues that may be applicable to other, similar, contexts; dependability (consistency of the findings) by exploring potential similarities among responses to two different questionnaires by two different groups of stakeholders; and lastly, confirmability (avoiding researcher bias), by ascertaining constant, ongoing, interaction among the authors of the project throughout the process.

Content analysis using ATLAS.ti software allowed the identification of 36 categories and 4 main themes emerging from the data. As noted above, minimal changes in the phrasing of the questions in both questionnaires facilitated the clustering of similar responses under the same themes and categories despite small nuances. This is why Figure 1 below only shows a single list of said themes and categories not separated by group. Some categories may seem similar because of how they are named (e.g., "better than expected" and "positive"), but they differ on the type of content allocated for each category (e.g., "better than expected" refers to content related to the expectations of student teachers and mentors in contrast to the reality they found, while the category "positive" refers to the pleasant or enjoyable feelings they had during the analyzed period). With all this in mind, the four main themes emerging from the data were:

- Attitudes and sensations: Effects of the pandemic on participants and students.
- Classroom management: Administrative and logistic issues inside/outside the classroom.
- Adaptations and restrictions: Resilience and modifications in classroom routines.
- Academic–experiential: Teaching pedagogy, including competences, methodologies, and/or development of educational materials.

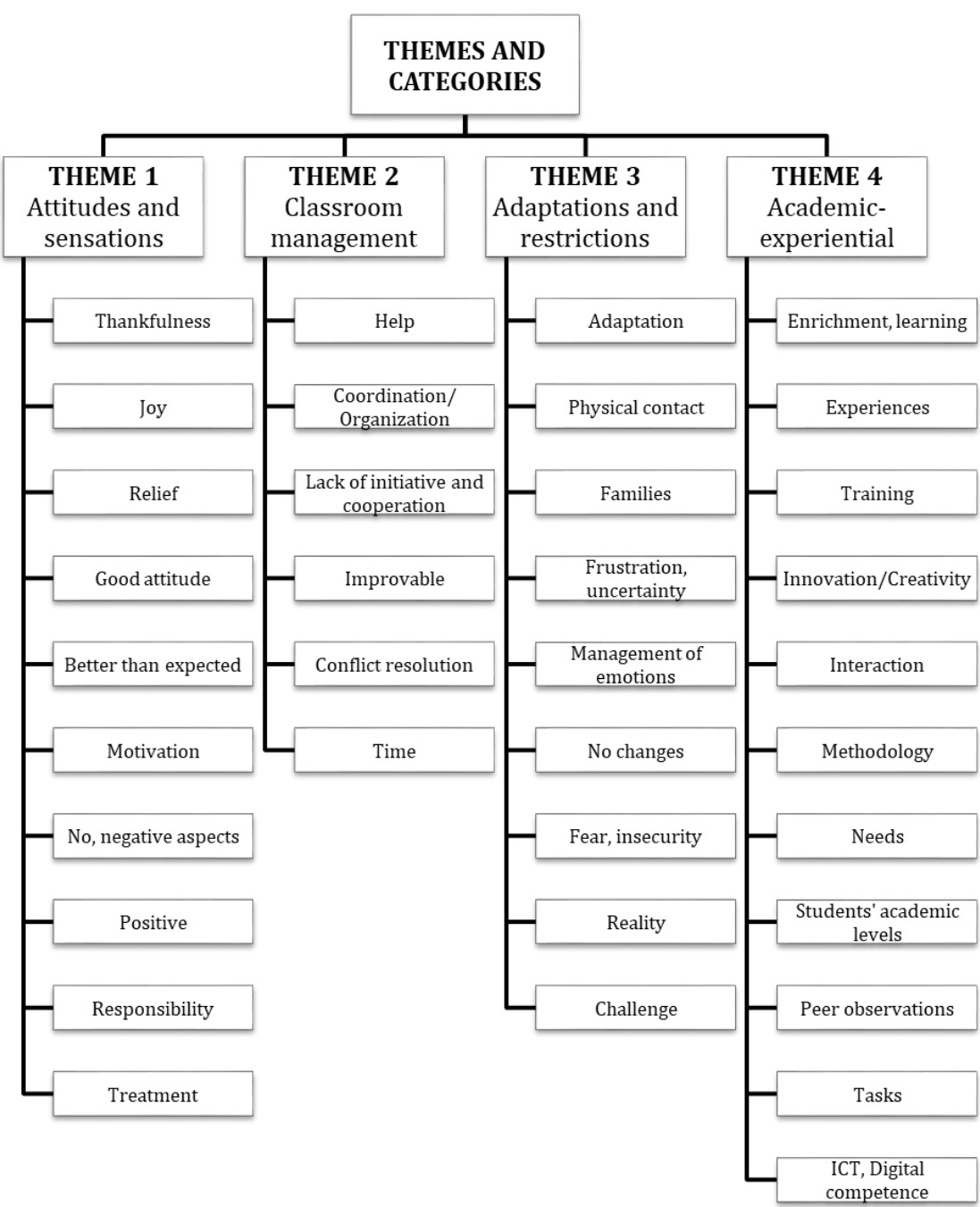

**Figure 1.** Themes and categories.

### 3. Findings

Figure 2 shows the interlocking nature of mentors' and student teachers' responses to the questionnaires, as well as the establishment of co-occurrent relationships among the themes emerging from the data. As can be seen, the number of student teachers' responses significantly surpassed that of mentors in the adaptations and restrictions and academic–experiential realms. Mentors' numbers, however, were larger when alluding to attitudes and sensations.

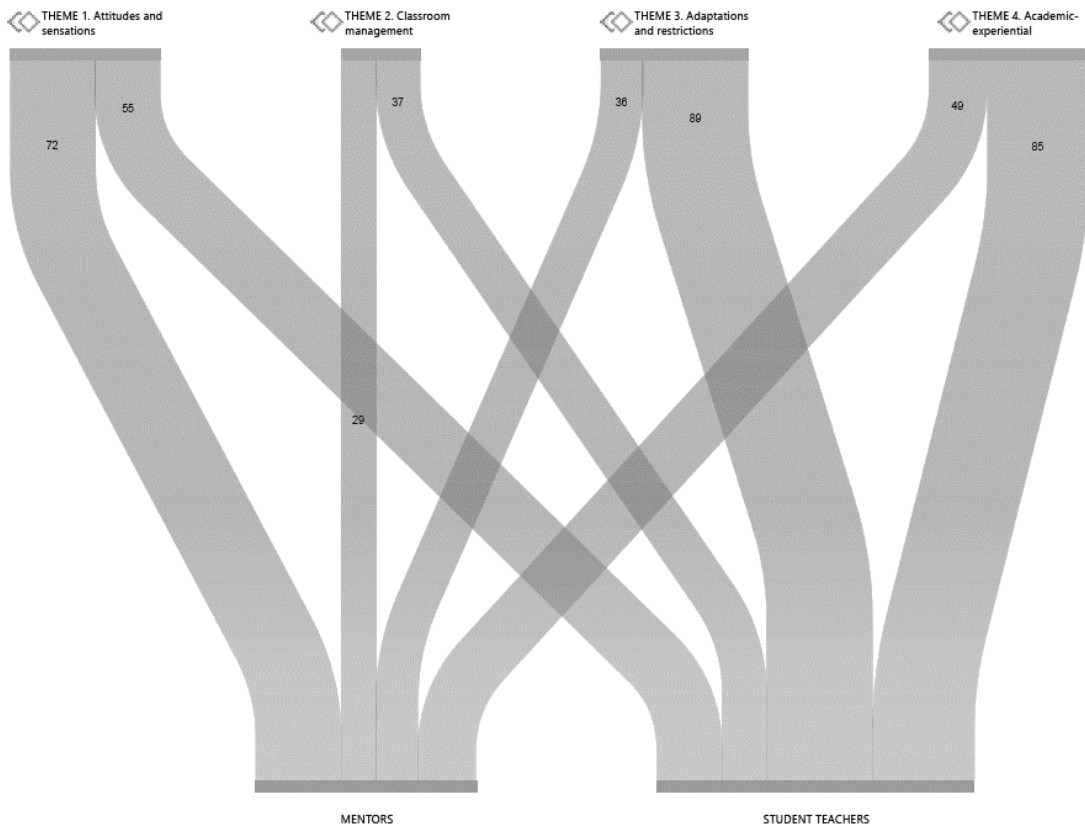

**Figure 2.** Frequency of responses.

Specific information about the number of student teachers' and mentors' responses within each theme and category is presented in double-entry Tables 2–5 in the following four subsections. Each column heading in the tables shows the letter I (Item), followed by the number of the corresponding questionnaire item (1 through 6), and by the initial of the stakeholder group (S = Student teacher, M = Mentor) providing the number of responses in the column. The rows in each table show the name of each category and the number of responses pertaining to them. Therefore, each cell where a column and a row intersect shows the number of respondents to each item and category for identification purposes. As an example, the numbers appearing in the I4S column in Table 2 indicate the number of student teachers' responses to item 4 in the questionnaire ("Broadly speaking, how would you evaluate your personal and professional contributions to the classroom during the pandemic?"), as well as the categories they pertained to (1—"Joy"; 2—"Better than expected"; 5—"Motivation"; 18—"Positive"). Similarly, column I4M shows the number of mentors' responses to the same item in their respective questionnaire ("Broadly speaking, how would you evaluate your assigned student teacher's personal and professional contributions to the classroom during the pandemic?"), as well as the categories clustering them (2—"Good attitude"; 1—"Motivation"; 22—"Positive"). Those individual student teachers' and mentors' quotes included in the narrative are cited verbatim in Spanish, followed by their English translation and the column and category they pertain to in parenthesis. For example, the following student teacher's explanation, "más fácil de lo que pensaba en un principio" [easier than what I thought at the beginning], in response to item 2 in the questionnaire ("How would you describe what student teaching has meant for you during the pandemic") is identified as (I2S-Better than expected) in the Attitudes and Sensations subsection below.

**Table 2.** Responses: attitudes and sensations.

| Theme | Category | I1S | I1M | I2S | I2M | I3S | I3M | I4S | I4M | I5S | I5M | I6S | I6M |
|---|---|---|---|---|---|---|---|---|---|---|---|---|---|
| **Attitudes and Sensations** | Thankfulness | | | 5 | 1 | | | | | | | 1 | |
| | Joy | | | 5 | | | 1 | 1 | | | | | |
| | Relief | | | | 3 | | | | | | | | |
| | Good attitude | | | | 1 | | 3 | | 2 | | | | 3 |
| | Better than expected | 1 | | 2 | 4 | | | | 2 | | | | |
| | Motivation | | 2 | | 1 | 1 | 1 | 5 | 1 | | 1 | | 1 |
| | No negative aspects | | | | | | | | | | 18 | | |
| | Positive | 1 | | 2 | 6 | | 1 | 18 | 22 | 10 | | 1 | |
| | Responsibility | | | | 1 | | 2 | | | | | | |
| | Treatment | | | 1 | | | 1 | | | 1 | 2 | | 1 |

**Table 3.** Responses: classroom management.

| Theme | Category | I1S | I1M | I2S | I2M | I3S | I3M | I4S | I4M | I5S | I5M | I6S | I6M |
|---|---|---|---|---|---|---|---|---|---|---|---|---|---|
| **Classroom management** | Help | | | | 12 | 11 | 4 | 4 | 4 | | | | |
| | Coordination/Organization | | | 1 | | 9 | 4 | | | 1 | | 3 | |
| | Lack of initiative and cooperation | | | | | | 1 | | 1 | | | | |
| | Improbable | | | | | | | | 1 | 1 | | | |
| | Conflict resolution | | | | | 3 | | | | | | | |
| | Time | | | 2 | | 2 | | | | 6 | | 1 | 1 |

**Table 4.** Responses: adaptations and restrictions.

| Theme | Category | I1S | I1M | I2S | I2M | I3S | I3M | I4S | I4M | I5S | I5M | I6S | I6M |
|---|---|---|---|---|---|---|---|---|---|---|---|---|---|
| **Adaptations and Restrictions** | Adaptation | 5 | | 8 | 1 | | | 3 | | | | | |
| | Physical contact | 25 | 14 | 3 | 2 | 3 | 3 | 3 | | 4 | 1 | | 1 |
| | Families | | | 2 | | | | | | | | | |
| | Frustration, uncertainty | 2 | 4 | 1 | 1 | | | | | 2 | 1 | | |
| | Management of emotions | | | 2 | | | 1 | 1 | | 1 | | | |
| | No changes | | | | 1 | | | | 1 | | | | |
| | Fear, insecurity | | | 4 | | | | | | 1 | 1 | | |
| | Reality | | 1 | 3 | 2 | | | 3 | | 5 | | 1 | 1 |
| | Challenge | | | 6 | | | | 2 | | | | | |

**Table 5.** Responses: academic–experiential.

| Theme | Category | I1S | I1M | I2S | I2M | I3S | I3M | I4S | I4M | I5S | I5M | I6S | I6M |
|---|---|---|---|---|---|---|---|---|---|---|---|---|---|
| Academic–Experiential | Enrichment and Learning | | | 7 | 3 | | | 3 | 1 | | | 2 | |
| | Experiences | | | 4 | 2 | | 1 | 2 | | 1 | | | |
| | Training | | | 5 | 1 | 1 | 1 | 3 | | | | 2 | 2 |
| | Innovation | | | 3 | 2 | 3 | 6 | | 1 | 1 | | | 2 |
| | Interaction | 1 | 4 | 1 | | 9 | 11 | | | | | 2 | |
| | Methodology | | 1 | 2 | 1 | 1 | | | | 5 | | 2 | |
| | Needs | | 2 | 1 | 1 | 1 | | 1 | | | | | |
| | Students' academic levels | | | | | | | | | 2 | | | |
| | Peer observations | | | | | | | | | 1 | | 2 | |
| | Tasks | 2 | 3 | | | 15 | 5 | 1 | 1 | 2 | | | |
| | Digital competence | | 2 | | | 15 | 9 | | | | | | |

### 3.1. Attitudes and Sensations

Table 2 shows the number of student teachers' and mentors' comments on aspects related to student teachers' presence in the classroom. Given the number of comments lauding the experience, both groups of stakeholders seemed to appreciate the opportunity to work together despite the existing health-related impositions and concerns.

As can be seen, three categories, "positive reactions to the experience", the "absence of negative aspects", and "motivation" elicited the largest number of responses across the questionnaire items. Item 4 in particular, which enquired on student teachers' positive contributions to the classroom, generated the largest number of responses from both student teachers (I4S = 18) and mentors (I4M = 22). The number of mentors' responses to item 5 (I5M = 18) denying potentially negative consequences of the pandemic on their relationship with student teachers and students, followed. Along the same lines, 10 student teachers (I5S) concurred with the mentors, highlighting the positive effects of the experience. This optimistic outlook was reflected in a student teacher's response to item 4 in the questionnaire (I4S-Motivation), which enquired on student teachers' evaluation of their contributions to the classroom. Her response revealed how, far from discouraging her, the adversities she faced pushed her to become a better practitioner:

> *Aunque me queda mucho por aprender y mejorar, considero que esta experiencia me ha hecho darme cuenta de mis puntos fuertes y débiles, de lo mucho que me gusta esta profesión y de lo mucho que quiero seguir mejorando, aprendiendo y cooperando con mis compañeros y alumnos para ser mejor maestra.*

> [Even though I still have a lot to learn and improve, I believe this experience has made me realize my strong and weak points, how much I love this profession and how much I want to continue to improve, learn, and cooperate with my colleagues and students to become a better teacher].

Mentors appreciated the "good attitude" of the student teachers, expressed "relief" upon being supported by them, and considered the experience "better than expected". Noticeably, 17 mentors complimented the student teachers when asked how they would describe what it meant for them to have a student teacher in class (I2M). Student teachers, for their part, were similarly appreciative of the opportunity to work with veteran educators and learn from them (I2S). The strong personal/professional connections developed during the practicum led one student teacher to refer to the latter as "más fácil de lo que pensaba en un principio" [easier than I thought at the beginning] (I2S: Better than expected). A

second one synthesized other fellow student teachers' opinions about the significance of the practicum, despite the extenuating circumstances surrounding it: "La mejor parte de la carrera y donde más aprendes es aquí, por lo que agradezco mucho que hayamos podido realizarlo" [the best part of the degree and where you learn the most is here, so I really appreciate that we have been able to carry it out] (I2S: Thankfulness).

*3.2. Classroom Management*

The figures in Table 3 reflect the number of student teachers' and mentors' responses to key aspects of student teachers' classroom management and logistical support, i.e., "help" with health-imposed protocols (I2M, I3S, I3M, I4S, I4M), "coordination and organization" of small and large groups during lessons (I1S, I2S, I3S, I3M, I5S, I6S), or keeping track of "time" (I1M, I3S, I5S, I6S, I6M). An overwhelming majority of comments were positive, save for a couple isolated exceptions raising issues about student teachers' "lack of initiative and cooperation" (I3M, I4M).

The two categories encompassing the largest number of responses were "help" and "coordination/organization", with 35 and 19 responses, respectively. Noticeably, 15 student teachers (I3S, I4S) and 20 mentors (I2M, I3M, I4M) concurred that student teachers had constituted an important source of help in the classroom. One student teacher, for example, explained that, "creo que he ayudado bastante a la tutora" [I believe I have helped my mentor a lot] (I3S-Help), while a second one noted that, "he ayudado en el desarrollo de las clases, al impartirlas, o bien prestando apoyo a todo el alumnado que lo necesitase" [I have helped in the development of classes, while teaching them, or while supporting all students in need] (I3S-Help). One of the mentors, for his part, praised her assigned student teacher for her support supervising students' individual work: "al trabajar con menor actividad grupal y dar más importancia a las actividades individuales, me ha resultado de gran ayuda" [in working with fewer group activities and giving more importance to individual activities, [she] has been a great help] (I3M-Help).

Several student teachers' responses in the "coordination/organization" category alluded to how the pandemic taught them to work closely with their mentors not just during the implementation of daily classroom routines, such as "mantener el orden y la limpieza" [maintaining order and cleanliness] (I3S: coordination/organization), but also while discussing pedagogically oriented issues: "las restricciones obligaron a adaptar las metodologías y a coordinarlas con los tutores" [restrictions forced us to adapt the methodologies and coordinate them with our mentors] (I3S: coordination/organization). In so doing, a student teacher learned a valuable lesson: "Cada clase tiene un ritmo diferente y considero que es importante saber cuál es el ritmo de cada una de ellas" [Each class has a different pace and I believe it important to know the rhythm of each one of them] (I6S: time). As for mentors, they agreed student teachers showed a general positive attitude while implementing and streamlining required health protocols, both inside and outside the classroom, and a genuine disposition to supervise students' work in their bubble groups: "organización de entradas y salidas del aula" [organization of coming in and out the classroom] or "organización del aula y desinfección" [classroom organization and disinfection], and also "al ser un grupo burbuja, la conexión con los alumnos es más estrecha" [as it is a bubble group, they have a closer connection with students] (I3M: coordination/organization).

Not everything appeared to run smoothly, though. In fact, a few student teachers voiced their discontent over incidents that tainted their relationships with their mentors or classmates. One of them, for instance, complained about "falta de comunicación fuera del horario escolar con el tutor" [lack of communication with my mentor outside the school schedule] (I6S: coordination/organization), while a second one lamented not having been able to join her friends during school breaks due to her considerable classroom duties: "me habría gustado poder salir en los recreos a descansar con mis compañeras" [I would have liked to go out during breaks to stop for a breath with my colleagues] (I6S-Time).

### 3.3. Adaptations and Restrictions

The rising number of contagions and deaths caused by the COVID-19 virus increased mentors' and student teachers' concerns about their own health. As can be seen in Table 4, 59 comments mentioned self-imposed restricted physical contact with students in their classrooms, and 11 and 6 comments alluded to feelings of "frustration/uncertainty" and "fear/insecurity", respectively. Given the constantly changing "reality" (16 comments) of their classrooms, both mentors and student teachers struggled to adapt to the new scenarios and manage their emotions.

Government-mandated school and classroom health protocols, such as wearing masks and keeping a safe physical distance from others, difficulted the delivery of academic instruction. One mentor, for example, brought up an evident challenge derived from having to wear a mask: "mantener la distancia y dar clases con las mascarillas dificulta el entendimiento" [keeping the distance and teaching classes with masks on hinders comprehension] (I1M: physical contact). Additionally, existing safety distance restrictions precluded mentors from placing students together in small or large groups. One student teacher, for instance, lamented "no poder realizar actividades grupales" [not being able to carry out group activities] (I5S: physical contact). For student teachers and mentors in lower elementary grades, the restrictions were especially disappointing, because they prevented them from having "la cercanía y el contacto físico con los alumnos y las familias" [the closeness and physical contact with students and their families] (I1S: physical contact, I1M: physical contact) they felt the youngest students needed.

As imposed physical contact restrictions began to be gradually lifted, student teachers and mentors had to learn to manage their emotions, despite overwhelming sensations of fear and frustration. In her response to what student teaching had meant for her during the pandemic (item 2 in the questionnaire), a student teacher explained that being in the classroom "ha supuesto miedo, porque la situación a la que nos enfrentábamos no era fácil" [It was frightening, because the situation we were facing was not easy] (I2S-Fear/Insecurity). Concurring with her view, a second one explained that, "me daba miedo contagiarme por exponerme tanto" [I was afraid of becoming contaminated for risking it so much] (I2S-Fear/Insecurity).

Given the continuously increasing number of COVID-19 cases, school and university administrators struggled to address unforeseen circumstances, such as finding replacements for mentors unexpectedly infected on a daily basis. Oftentimes, finding qualified personnel was an unsurmountable task, due to the limited pool of candidates. However, student teachers did not seem to understand this plight. One of them, for example, denounced "la poca organización y seriedad en la contratación del profesorado [universitario], ya que hemos tenido cuatro tutores en el período de prácticas" [the poor organization and seriousness in the hiring of [university] professors, as we have had four mentors during student teaching] (I1S: frustration/uncertainty). Ongoing alterations in regular school routines, such as changes and modifications in daily schedules, academic goals, or initial objectives frustrated one mentor, who explained that, "tenía unas expectativas sobre innovación educativa y pedagogías emergentes que me han sobrepasado, ya que no sabía lo complicado que podía ser aplicar todo lo que me gustaría" [I had a few expectations about educational innovation and emergent pedagogies that have surpassed me, because I didn't know how complicated it could be to apply everything I would like] (I1M: frustration/uncertainty). The merger of the factors in this subsection took a toll not just on student teachers and mentors, but also on the students themselves, those on the receiving end of adaptations to changing directives on an almost daily basis. As one mentor (I1M: reality) explained, many of them began to show signs of distress and lack of motivation in the classroom that also affected their teachers:

> *Alguna vez me he sentido abrumado porque no he sabido abordar adecuadamente una actividad o un tema, porque el alumnado se encontraba un poco más cansado y desmotivado y porque, a pesar de tener poco alumnado, es más difícil de lo que en un principio podía pensar.*

[I have felt overwhelmed on occasion because I haven't known how to approach an activity or topic adequately, because students were a bit more tired and unmotivated and because, despite having only a few students, it is more difficult than I thought in the beginning.]

*3.4. Academic–Experiential*

The academic–experiential theme encompassed the categories in Table 5. Help with tasks (29), interaction (28), digital competence (26), innovation (18), and enrichment and learning (16) were the categories compiling the most comments.

Student teachers valued the practicum as a very enriching period of time, because it allowed them to see firsthand the realities of the classroom from a different standpoint, interact with students and veteran educators, and acquire valuable know-how for their own future as educators. In fact, they described the practicum in their comments as "pleasurable", "beautiful", or "unique". One student teacher highlighted its educational significance as, "las prácticas, la mejor asignatura y donde más aprendemos" [student teaching, the best subject and where we learn the most] (I2S: enrichment and learning).

Mentors, for their part, appreciated their respective mentees' leadership qualities while organizing physical tasks ("realizó actividades y juegos para la parte de psicomotricidad" [She carried out activities and games during psychomotricity time]) (I3M-Tasks), or technology sessions ("Desarrolló sesiones a través de la plataforma digital" [He developed sessions via the digital platform]) (I3M: digital competence). They also acknowledged student teachers' dedication to connect with students despite the impact of the health restrictions imposed. One mentor, for instance, explained that "se muestra muy cercana a los alumnos e interactúa con ellos en todo momento" [she is very close to the students and interacts with them at all times] (I3M: interaction). Student teachers, for their part, were generally more specific when explaining their precise contributions to the classroom. Thus, one of them explained that he helped students perform tasks during small group work ("ayuda en los pequeños grupos-burbuja a la hora de realización de tareas" [help small bubble groups at the time of completing tasks]) (I3S: tasks), while a second one listed examples of activities he introduced to increase students' participation in class ("he incorporado actividades educativas activas, la psicomotricidad y juegos" [I incorporated active educational activities, psychomotricity, and games]) (I3S: innovation). Notwithstanding the overwhelming presence of positive comments, one student teacher missed the opportunity to have observed other mentors in their classrooms due to then-strict limitations in place: "…no poder observar cómo trabajan en más clases o maestros debido a que no se puede entrar en otras aulas por el COVID" [not to be able to observe how [they] work in more classes or teachers given that we cannot enter other classrooms because of the COVID] (I5S: methodology).

Both mentors and student teachers seemed noticeably concerned about students' emotional wellbeing and strived to help the latter feel as comfortable as possible in class. One mentor, for example, explained that "para mí, ha sido prioritario lo que le sucedía a cada niño . . . hemos sido animadores para los niños desde el punto de vista profesional" [for me, my priority was what happened to each child... we have been cheerleaders for the children from the professional standpoint] (I1M: needs). However, this legitimate concern did not sidestep mentors or student teachers' focus on the academic component of their job. Working closely together, they adapted, or searched for, innovative pedagogical methods and technological tools to provide instruction and foster student participation in class. Student teachers' leadership and initiative were invaluable in this regard, as their familiarity with apps and technological devices introduced resources in the classroom such as Genially, Kahoot, ClassDojo, or Plickers. Moreover, they were able to expand their mentors' knowledge of resources already present in class. One student teacher, for example, explained that, "creo que he ayudado mucho a mi tutora con la pizarra digital, ya que me gusta mucho investigar y ver qué podemos sacar de las nuevas tecnologías y cómo emplearlas en el aula" [I believe I helped my mentor a lot with the digital board, because I

really like to investigate and see what we can do with the new technologies and how to use them in the classroom] (I4S: training). Mentors were very appreciative of this crucial support. Despite several of them having achieved high levels of digital competence per the European Framework for the Digital Competence of Educators [41], as shown in Figure 3, they were not reticent to recognize their limitations. One of them, for instance, openly admitted that her student teacher "posee más conocimientos que yo" ([he] possesses more knowledge than I) (I2M: enrichment and learning), while a second one synthesized what seemed to be the general opinion among his colleagues about the impact of student teachers' technological expertise on lessons and presentations: "hoy en día estos jóvenes maestros vienen muy bien preparados en TIC y la presentación de las propuestas para presentar los contenidos de cada una de las materias se deja notar" [Nowadays, these young teachers are very well prepared in information and communication technologies (ICTs) and the presentation of the proposals to present the content in each subject is noteworthy] (I3M: digital competence).

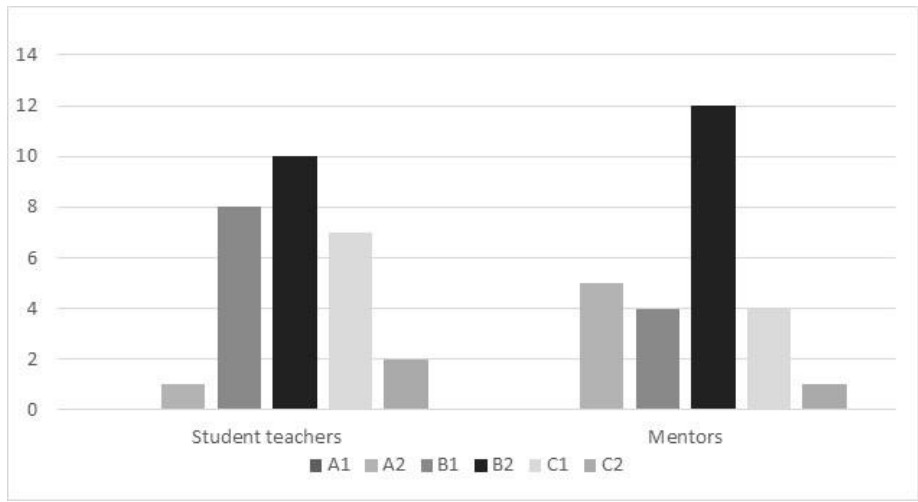

**Figure 3.** Mentors and student teachers' digital competence levels.

Overall, the practicum was perceived as an enlightening, mind-opening, and immense learning experience for all. For one student teacher, it became "lo más bonito de la carrera" (the most beautiful [component] of the degree) (I2S: experiences), in that "hemos aprendido a dar lo mejor de nosotros como docentes" [We have learned to give the best of us as teachers] (I2S: experiences). For her, and some of her fellows, the challenges they underwent in the classroom during their return to face-to-face instruction boosted their intrinsic motivation [42] to become better professionals [43]. As for mentors, student teachers' predisposition to help, individual initiative, forward-looking attitude, and concern for students, helped keep everyone afloat during a tragic period in their personal and professional lives. The following quote by one mentor (I6M: training) summarized his overall perception of student teachers' contributions during the practicum:

> *Los alumnos de prácticas normalmente se involucran mucho, aportan ideas nuevas y esto les pone en contacto con el mundo tan fascinante de la infancia. Creo que hacemos posible que se vayan con ilusión para dedicarse a esta preciosa profesión.*

> [Student teachers are normally deeply involved, contribute new ideas and this puts them in touch with the fascinating world of childhood. I believe we make it possible they leave happy to devote themselves to this beautiful profession.]

## 4. Discussions

In the middle of an uncontrollable pandemic that disturbed the personal and professional lives of so many educators and students worldwide [3,31,44], educators around

the world strived to offer students high quality education via remote modes of instruction, hybrid classrooms, and in-person lessons following the reopening of schools. In the present study, the return to face-to face instruction for UEx mentors and student teachers meant the introduction and adaptation of new and/or existing teaching methods and technological tools, as well as logistical classroom changes and reorganizations pending the evolution of the virus, all the while facing their own, and their students', fears and uncertainties [45]. Despite conflicting, and sometimes unclear, health-related directives related to their physical presence in the classroom, student teachers regarded the practicum as the best component of their degree, due to the myriad of learning opportunities to grow personally and professionally. Working closely with their mentors and observing the students in their classrooms, they developed resilience, compassion, and empathy, among many other invaluable life skills [46]. Their comments reporting increases in their motivation to become good educators, willingness to help their mentors, especially in the digital realm, and interest in the wellbeing of students showed that it is possible to achieve reflexivity and critical learning during internships [47].

The decision to reopen schools during the pandemic carried with it a tremendous amount of responsibility for mentors and student teachers, as they were forced to implement, and comply with, continuously changing health and safety protocols. As a result, classroom management and administrative and teaching decision-making procedures became inherent, and unanticipated, practicum companions. Similarly to thousands of worldwide classrooms and schools, the sudden surge of the virus transformed participants' classrooms, traditionally a space for formative and experiential enrichment [48], into experimental laboratories, subject to hastily developed, promptly released, academic, administrative, and socio-educational decrees [49]. Fortunately, mentors and student teachers internalized their roles as resourceful agents of change, pulling their respective weight to make students' educational experiences more bearable during a very difficult time. Their laudable concern for the well-being of their students led them to attempt to transform their classrooms into the safe spaces students needed to escape the existing chaos outside and concentrate, if at all possible, on the academic content they needed to learn to achieve their respective grade-level objectives.

A positive effect of the outbreak of the pandemic was the introduction and adaptation into the classroom of instructional methods and digital tools that partially replaced more traditional methods of teaching. Mentors and student teachers alluded to the presence in their classes of more innovative, interactive practices, focused on collaborative work, that increased student engagement and participation through the use of software apps, and created spaces for dialogue and autonomy [50]. The leadership of the student teachers played a critical role, because they were able to design activities revolving around the use of interactive apps that made lessons more appealing. At the same time, they were also able to help their mentors become more familiar with applications of existing tech tools in the classroom the latter might not have been aware of until then. Interestingly, as noted in the academic–experiential subsection, although all the mentors had achieved different digital competence levels, they recognized their limitations and the superior preparation of the student teachers in the technological field, thereby showing that being in possession of a certificate does not necessarily guarantee teachers will make ample use of technology to deliver content [51].

## 5. Conclusions

The dramatic spread of COVID-19 affected educational institutions worldwide [3]. Thousands of schools closed their doors only to progressively reopen them subject to drastic face-to-face restrictions and health-related measures [4,5]. The present project expanded on the results of a pilot study by three of its coauthors [52], analyzing the impact of the pandemic on mentors and student teachers in the Early Childhood and Elementary Education degree program at the University of Extremadura during the practicum. Concurrent with the results of said study, the findings of the present project revealed the participants'

positive take on the experience, despite serious concerns about their own health and safety. These findings, though, are limited to the opinions of the participants in the study. Other student teachers and mentors in the same or other degree programs might have held different opinions about the practicum and their participation in it.

Up until now, the scientific literature has revolved around the beneficial effects of the practicum on student teachers, with a special emphasis on the know-how acquired from their mentors [47,53]. The exceptional circumstances surrounding the pandemic, however, partially reversed the outcomes abovementioned, making teachers coparticipants and even leaders, given their expertise in certain instructional areas. Their extensive familiarity with digital technologies, for example, turned them into experts, able to help mentors improve teaching practices and develop engaging activities that somewhat relieved the latter, as well as the students, from the fear, fatigue, and anxiety generated by the pandemic. Simultaneously, their positive interaction with students created a more calming social–emotional learning environment [54]. With this in mind, educational institutions and fieldwork supervisors might consider publicizing the results of successful mentor–student-teacher collaborative efforts during the practicum, perhaps in the form of a "good practices" catalog, that can inspire newer student teachers to bring innovative practices to the classroom while learning from more seasoned educators. For Allen [55], continuous practice and support fosters teachers' sense of belonging in the profession. As shown in this study, student teachers' contributions and dexterity with innovative instructional tools constituted an extraordinary source of help in the middle of a debacle significantly altering the lives of so many.

**Author Contributions:** Authors contributed equally to this work. All authors have read and agreed to the published version of the manuscript.

**Funding:** The publication of this work has been possible thanks to the funding granted by the European Regional Development Fund (FEDER) of the European Union and by the Junta de Extremadura (Ministry of Economy, Science and Digital Agenda), "Ayuda a Grupos GR21141". This grant has been co-financed by FEDER funds, FEDER operational programme of Extremadura.

**Institutional Review Board Statement:** The study was conducted in accordance with the Declaration of Helsinki, and following the considerations of Ethics Committee of the University of Extremadura and using as a reference the consent form informed provided by the Committee.

**Informed Consent Statement:** Informed consent was obtained from all subjects involved in the study.

**Data Availability Statement:** Not applicable.

**Conflicts of Interest:** The authors declare no conflict of interest.

## Appendix A

| Questionnaire for student teachers |
| --- |

**Demographic data:**
(a) Grade and major.
(b) Digital competence level per the European Framework for the Digital Competence of Educators.
(c) Type of school placement: public/semi-pubic/rural.

**Open-ended questions:**
**1.** As a student teacher at your placement school, what were your main challenges during the pandemic?
**2.** How would you describe what student teaching has meant for you during the pandemic? Why?
**3.** What were your main contributions to your classroom placement during the pandemic? Please include specific examples (bringing in digital tech tools, classroom organization, interaction with students...)?
**4.** Broadly speaking, how would you evaluate your personal and professional contributions to the classroom during the pandemic?
**5.** Were there any negative aspects of the pandemic you would like to comment on?
**6.** Feel free to add any other considerations not addressed in the questions above.

**Appendix B**

| **Questionnaire for mentors** |
| --- |
| **Demographic data:**<br>**(a)** Grade taught, specialization, and years of experience as a teacher and as a mentor teacher.<br>**(b)** Digital competence level per the European Framework for the Digital Competence of Educators.<br>**(c)** Type of school where you teach: public/semi-public/rural.<br><br>**Open-ended questions:**<br>**1.** As a mentor teacher at your school, what were your main challenges during the pandemic?<br>**2.** What did it mean for you to have a student teacher in your class during the pandemic? Why?<br>**3.** What were your assigned student teacher's main contributions to the classroom during the pandemic? Please include specific examples (bringing in digital tech tools, classroom organization, interaction with students…)<br>**4.** Broadly speaking, how would you evaluate your assigned student teacher's personal and professional contributions to the classroom during the pandemic?<br>**5.** Were there any negative aspects of the pandemic you would like to comment on?<br>**6.** Feel free to add any other considerations not addressed in the questions above. |

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
