# Peer review of "Student Teachers’ Classroom Impact during Their Practicum in the Times of the Pandemic"

_education, doi:10.3390/educsci13030277_

Round 1
Reviewer 1 Report
Thank you for the opportunity to review the paper. Overall, the article covers an interesting and important topic. It is particularly positive to mention that the study also asked about and elaborated on the student teachers' contribution to improving the situation in schools. Nevertheless, in my opinion, some changes are still necessary before the publication of the article, which I would like to elaborate on in the following.
Abstract:
Please specify the research question or the research objective and the methodological approach of the study.
Introduction:
Please support the complete second paragraph (lines 25 - 33) of the introduction with appropriate literature.
Chapter 1.1:
Please provide further details on how long the schools were closed and how schools delt with the need to provide distance education. In my opinion you can also shorten the list of institutions and buildings that were also closed due to more space for relevant institutions.
The description of remote teaching in line 53 raises an issue that should already have been mentioned in chapter 1.1, since the need to switch to digital teaching already existed there. Maybe you could relate both paragraphs more to each other and introduce the term "remote teaching" right in paragraph 1.1 as schools had to provide lessons during the closures as well as after the reopening.
line 63: "Moreover, other reports evidenced the lack of preparedness of the educational system to support teachers’ and families’ technological needs during this time [12]." You should elaborate more on teachers digital readiness before this statement. Your previous sentences are only concerned with families and their technical equipment. Thus your remark on teachers is not very well prepared and also not embedded in existing research.
Chapter 1.3 (last paragraph):
You should spent some more words on the changes students and mentors had to cope with during school closures and the reopening of schools. Your research is very interesting and valuable but I would also recommend that you formulate at least a research question and elaborate on the added value and the aim of your study in detail. There is also some existing research on practical experiences of teacher students during the pandemia. It is also not really clear if you examine practica during school closures or pracitca after the reopening of schools in this paragraph. It gets a little more clear in the following but should be mentioned right before the methods-section.
2. Methodology:
I would recommend that you start the description of the method with a subchapter on study design, since it is not clear which methodological approach you choose. There you should describe the chosen approach and argue why you have chosen it and where the added value lies. In addition, you could move the subchapter 2.3 here. It is difficult to assess the fit of the method to the research objective, as the objective or the reason for choosing the method is not clearly stated. To improve the reader friendliness I would recommend that you sum up the aim and the added value of the study before describing the method.
Table 1: All abbreviations in the table (e.g. the ending "Ed") should be explained. Please provide also more information on the selection process of teacher students and mentors.
2.2 Instruments: Please add some informations on the sociodemographic aspects captured within the questionnaire.
Figure 1: Some of the subcategories seem to be not distinct e.g. "better than expected" and "positive". "positive" could also be seen as a generic term for most of the other terms. Please describe in more detail how the subcategories differ from each other. There is also a misspelling of 'metodology'. Tt should be 'methodology'.
3. Findings:
Overall, it is noticeable that the reference to the table is often missing in the text and the table is inconveniently placed. It gives the impression that the table is in the middle of the description of the results. Move the table either to the beginning or the end of the subchapters and introduce the table better in the text as you did in subchapter 3.4.
It would make it much easier for the reader to understand the formation of the categories and the classification of the statements into them if you were to show the coding (e.g., using the example of I5S in chapter 3.1) or if you were to give an anchor example for each category. This is a common procedure in the presentation of category systems and is highly recommended by me. Although some examples follow in the text, they often cannot be clearly assigned to the subcategories.
You really often refer to statements made by respondents without assigning them to subcategories (e.g., Lines 282 - 287). Please make the assignment of the examples to the subcategories clear in the text.
Figure 2: The figure is innovative, but the added value of the figure is not clear. What is the added value of this figure? What contributes the sheer frequence of mentioned aspects to the research aim? The reference to the fact that one category seems to be more important for the students and the other category more important for the mentors is interesting, but should be further substantiated by the direction of the mentions. So It would be interesting not only to mention the frequency of responses but also wether they were positive or negative. Additionally, the different weighting of the areas by students and mentors should be revisited in the discussion if it is so prominent in the results section.
Table 2: Am I right that the 'I' refers to interview, the number refers to participand, and 'S' or 'M' refers to student or mentor? If so, it is a bit surprising that one person made 16 or more mentions to one category. Or does the 'I' refer to the interviewer and his interpretation? This would fit better with the frequencies, but cannot be clearly read from the table and is also not explained in the notes to the table. The following paragraph however, makes it clear that the abbreviations refer to partipants. Please provide further informations on the abbreviations and add notes to the table. Again, 22 mentions of one category seem quite much for one participant. Please elaborate on this aspect. There is also one blank too many at 'I1 S'.
Table 3: According to APA guidelines for manuscript preparation horizontal lines should be displayed in tables as little as possible. Please design the tables in a uniform manner according to the APA guidelines.
Chapter 3.2:
This chapter is better than the previous one because of the examples of one subcategory. But you go into great detail about the coordination/organization subcategory, but completely neglect the other categories. Explain this approach better. Alternatively, you should provide anchor examples for all subcategories. In addition, I think the table is in an inappropriate place in the middle of the text. Move the table to the end or the beginning of the chapter and refer to the table in the text.
Chapter 3.3: Please insert the assignment of the examples to the subcategories and refer to the table in the text.
Lines 305 - 307: Was the assignment to topic 3 certainly made because the question was directly related to the topic? At first glance, no top category from Theme 3 seems to fit into Figure 1, since organizational aspects do not seem to be covered anywhere there. Please explain why no extra top category was formed here if statements were obviously made in this direction. In general, examples should again be provided for all subcategories. Alternatively, only the categories with the most mentions can be provided with examples, since the word count probably does not allow for more detail. However, this should be announced and justified in the text.
Chapter 3.4: The division of the text before and after the table is well done. Before the table, the different categories are introduced and referred to, followed by more detailed explanations and examples after the table. I would recommend this for all chapters.
4. Discussion:
You should mention in a few sentences the different weighting of the categories with regard to the frequency with which they are mentioned by students and mentors, as this is quite prominent in the results section due to figure 2.
typo in line 418: 'remotes' instead of 'remote'
5. Conclusion: The studies mentioned in line 459 and following lines should be mentioned already in the theoretical background. You could reference existing assessments of internships there, then transit to existing findings on practica during the pandemic and work out your research desideratum from there. This is missing, as is the formulation of a concrete research goal.
Author Response
Please see the attachment to review the modifications suggested by the reviewers.
Thanks.

Reviewer 2 Report
The goal of the paper is to summarize the experiences of the student teachers and the mentors at the University of Extremadura in connection with the extraordinaire circumstances in the education during the COVID pandemic.
The questions were in questionnaires, and the answers are texts. These texts are analyzed by the software ATLAS.ti. The answers might be very different as the number of groups are quite large in each theme.
Definitely, it is a case study. I do not understand why the researchers made sample containing 28 people. Why were not all student teachers asked? If a software analyses the data, the analysis would not have been too difficult, and more information would have been gained.
If I understand well, the survey was made during the reopening period. The questions concern the lockdown or the reopening period? It is not totally clear, because the questions use ”during the pandemic” expression. Do the questions concern the hybrid education?
The paper contains lots of information which are not important (for example “A foreign tourist in the Canary Islands became the first official COVID case in Spain 35 in January of 2020”, “While not awarded monetary stipends for their work with student teachers, they are granted additional points should they decide to request a transfer to another school)” and so on.
The conclusion: Student teachers could help the mentors. I agree with this statement. It presumably due to the better skills in informatics (which might be connected to the age factor). I think that if the teacher (not student teacher) was young, he/she does not require so much help. It could be analyzed, as well trough the digital competence level. In general, although some demographic data are asked, no comparison was made using them. If more respondents were in the survey, these analyses could be done.
Summarizing my opinion: the paper does not contain important contributions. It is too large and is not clear with respect to some views. It does not have enough respondents to make not-descriptive statistics.
Author Response

(The authors gave the same response as above.)

Round 2
Reviewer 1 Report
You have revised the manuscript extensively, addressing a number of my comments. The statistical procedure and the aim of the research are now well elaborated and supported by the findings, which are now presented coherently in the results section. The discussion has also gained substance. Of course, this is a fairly small sample, but in my judgment appropriate to the method chosen (document analysis). In my opinion, the manuscript has gained in quality, so that it can now be published in its present form.
Reviewer 2 Report
It has been significantly improved.
The supplementary material is not available on the website www.mdpi.com/xxx/s1
Please upload if it exists.